# Characterization of caffeine response regulatory variants in vascular endothelial cells

Carly Boye[1], Cynthia A Kalita[1], Anthony S Findley[1], Adnan Alazizi[1], Julong Wei[1], Xiaoquan Wen[2], Roger Pique-Regi[1,3]*, Francesca Luca[1,3,4]*

[1]Center for Molecular Medicine and Genetics, Wayne State University, Detroit, United States; [2]Department of Biostatistics, University of Michigan, Ann Arbor, United States; [3]Department of Obstetrics and Gynecology, Wayne State University, Detroit, United States; [4]Department of Biology, University of Rome Tor Vergata, Rome, Italy

**Abstract** Genetic variants in gene regulatory sequences can modify gene expression and mediate the molecular response to environmental stimuli. In addition, genotype–environment interactions (GxE) contribute to complex traits such as cardiovascular disease. Caffeine is the most widely consumed stimulant and is known to produce a vascular response. To investigate GxE for caffeine, we treated vascular endothelial cells with caffeine and used a massively parallel reporter assay to measure allelic effects on gene regulation for over 43,000 genetic variants. We identified 665 variants with allelic effects on gene regulation and 6 variants that regulate the gene expression response to caffeine (GxE, false discovery rate [FDR] < 5%). When overlapping our GxE results with expression quantitative trait loci colocalized with coronary artery disease and hypertension, we dissected their regulatory mechanisms and showed a modulatory role for caffeine. Our results demonstrate that massively parallel reporter assay is a powerful approach to identify and molecularly characterize GxE in the specific context of caffeine consumption.

*For correspondence:
rpique@wayne.edu (RP-R);
fluca@wayne.edu (FL)

Competing interest: The authors declare that no competing interests exist.

## Editor's evaluation

This important study identifies context-specific regulatory variants by an MPRA screen in vascular endothelial cells exposed to caffeine. The authors use a compelling and creative approach to pinpoint potential molecular mechanisms of gene-by-environment effects on gene regulation. The variants they identify are likely linked to complex disease risk.

## Introduction

Caffeine is the most widely consumed stimulant in the world (***Planning Committee for a Workshop on Potential Health Hazards Associated with Consumption of Caffeine in Food and Dietary Supplements, Food and Nutrition Board, Board on Health Sciences Policy, Institute of Medicine, 2014***). Caffeine produces a vascular response in the endothelium, causing vasodilation. The vascular endothelium, the innermost layer of arteries, is involved in several important functions, including regulation of blood flow, angiogenesis, thrombosis, and coagulation (***Hadi et al., 2005***; ***Krüger-Genge et al., 2019***). Endothelial dysfunction occurs in diseases such as atherosclerosis and hypertension (***Xu et al., 2021***), eventually leading to coronary artery disease (CAD) (***Matsuzawa and Lerman, 2014***). Multiple studies have investigated the role of caffeine in cardiovascular disease (CVD), and more broadly, vascular health in general, with conflicting results (***Chieng et al., 2022***; ***Ding et al., 2014***; ***Turnbull et al., 2017***) on the role of caffeine in CVD risk. ***Ding et al., 2014*** meta-analyzed

36 studies and found no association between heavy coffee consumption and increased risk of CVD (*Ding et al., 2014*). Similarly, *Turnbull et al., 2017* observed that moderate caffeine consumption was not associated with an increased risk of CVD or other cardiovascular events such as heart failure (*Turnbull et al., 2017*). Multiple studies suggested that caffeine may be beneficial in reducing the risk of CAD (*Choi et al., 2015*; *Miranda et al., 2018*; *Voskoboinik et al., 2019*), while others provided evidence that caffeine may reduce the risk of heart failure, but had no significant effect on the risk of coronary heart disease or CVD (*Stevens et al., 2021*). Most recently, *Chieng et al., 2022* found that decaffeinated, ground, and instant coffee significantly decreased CVD risk and mortality (*Chieng et al., 2022*). The conflicting results from these epidemiological studies may have several causes, including potential interactions between caffeine consumption and other environmental and genetic risk factors. Recent molecular studies investigated the consequences of caffeine exposure on chromatin accessibility and gene expression in vascular endothelial cells (*Findley et al., 2019*). This study identified response factor motifs for caffeine, defined as transcription factor motifs that are enriched in differentially accessible regions, and demonstrated that caffeine can induce changes in gene regulation in endothelial cells.

Analyzing the changes in gene expression upon exposure to environmental stimuli is a powerful approach to discover genotype–environment interactions (GxE). These molecular GxE result in a different response depending on genotype (*Knowles et al., 2018*; *Knowles et al., 2017*; *Moyerbrailean et al., 2016b*), potentially through allele-specific effects (ASE) on response factor binding or other environmental-specific gene regulatory mechanisms. Yet regulatory sequences that are differentially bound in response to environmental perturbations are poorly annotated. Single-nucleotide polymorphisms (SNPs) within caffeine response factor binding sites were enriched for artery expression quantitative trait loci (eQTLs) colocalized with CAD risk variants (*Findley et al., 2019*). The results of this study thus suggested that SNPs within regulatory elements active in the presence of caffeine may play a role in CAD risk and pointed to GxE in gene regulation as a potential mechanism underlying caffeine modulation of genetic risk for CAD. However, only a limited number of molecular GxE for caffeine have been studied so far, thus the transcription factors and regulatory sequences involved in caffeine GxE remain uncharacterized. Furthermore, it is important to study GxE in the relevant cell type; that is, endothelial cells which constitute the vascular endothelium. For these reasons, it is crucial to investigate and validate the mechanisms behind caffeine GxE in vascular endothelial cells.

Massively parallel reporter assays (MPRA) have allowed studies of noncoding genetic variants and their role in gene regulation, at unprecedented scale (*Arnold et al., 2013*; *Gordon et al., 2020*; *Kalita et al., 2018*; *Melnikov et al., 2012*; *Patwardhan et al., 2012*; *Tewhey et al., 2016*; *Ulirsch et al., 2016*; *Vockley et al., 2015*; *Wang et al., 2018*). Originally developed to study the gene regulatory potential of promoters and enhancer sequences, MPRA protocols have been further developed to study regulatory genetic variation and fine map association signals (*Kalita et al., 2018*; *Tewhey et al., 2016*; *Ulirsch et al., 2016*; *Vockley et al., 2015*). MPRAs with synthetic regulatory sequences can test allelic activity for candidate regulatory variants independently of their allele frequency in the population (*Kalita et al., 2018*; *Tewhey et al., 2016*; *Ulirsch et al., 2016*; *Vockley et al., 2015*). In MPRAs, DNA sequences containing each allele are transfected into cells and RNA-seq is used to quantify the transcripts for each allele. To directly test allelic effects of tens of thousands of candidate regulatory sequences predicted to affect transcription factor binding (CentiSNPs; *Moyerbrailean et al., 2016a*), we used an MPRA called Biallelic Targeted STARR-Seq (BiT-STARR-Seq) (*Kalita et al., 2018*). Only two previous studies have used MPRAs to investigate DNA sequences that regulate the transcriptional response to treatments (*Johnson et al., 2018*; *Shlyueva et al., 2014*). One study utilized STARR-Seq to characterize enhancer activity in *Drosophila* cells upon treatment with the hormone ecdysone; however, it did not investigate GxE (*Shlyueva et al., 2014*). The other study utilized STARR-Seq to investigate the response to glucocorticoid treatment in the human cell line A549 (*Johnson et al., 2018*). Although this study investigated GxE interactions, only a small number of variants were tested as this study was limited to preexisting variation within the samples (as opposed to designed target sequences) and only two variants had significant GxE (*Johnson et al., 2018*). We aim to identify and validate the DNA sequences that regulate the transcriptional response to caffeine in the vascular endothelium and how genetic variation present in these regulatory elements may affect the transcriptional response to caffeine (*Figure 1A*).

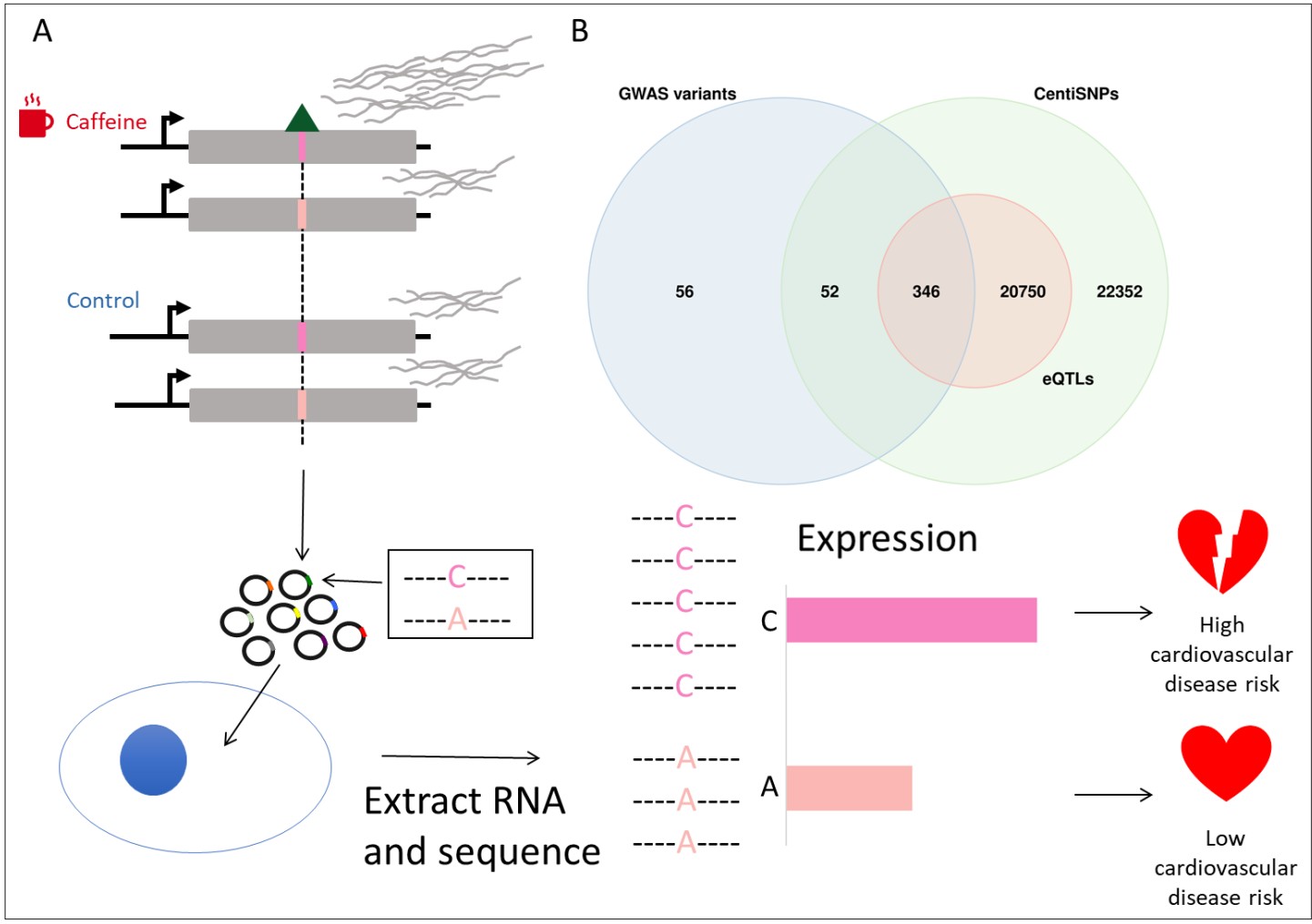

**Figure 1.** Study design. (**A**) Genetic variants modulate transcriptional response dependent on environmental conditions. The pink bars represent different alleles present in our targets, and the green triangle represents a bound transcription factor. These target sequences are transfected into cells, RNA is extracted and sequenced, and then activity is measured for targets for both alleles. (**B**) Library composition based on annotation category: single-nucleotide polymorphisms (SNPs) predicted to alter transcription factor binding using a combination of sequence information and experimental data (centiSNPs) (*Moyerbrailean et al., 2016b*), SNPs associated with complex traits (genome-wide association studies [GWAS]), and expression quantitative trait loci (eQTL) in GTEx.

The online version of this article includes the following figure supplement(s) for figure 1:

**Figure supplement 1.** Coverage histograms for all 12 libraries used in this study.

## Results

### Active regulatory regions in response to caffeine

For this study, we used a library of targets that was designed to capture a large number of predicted gene regulatory variants in motifs for hundreds of transcription factor binding sites (*Figure 1B*). The targets in our library consisted of self-transcribing enhancer regions containing a minimal promoter, a reporter gene, and the oligonucleotide containing the candidate regulatory SNP. These candidate regulatory SNPs belong to multiple categories, including SNPs predicted to alter transcription factor binding using a combination of sequence information and experimental data (centiSNPs) (*Moyerbrailean et al., 2016b*), SNPs associated with complex traits (genome-wide association studies [GWAS]), and eQTLs in GTEx. In addition, we included 1676 negative control sequences. To test if these putative regulatory sequences mediate the response to caffeine, we used DESeq2 to test for differential activity of the targets in cells treated with caffeine compared to cells in the control group (see 'Materials and methods' for the specific model). The library contained motifs in both the forward

and reverse orientations within separate targets. Since these motifs may induce direction-specific effects, we performed the differential activity analysis per each direction separately (see *Figure 2—figure supplement 1*; see also *Figure 2—figure supplements 2 and 3*), and considered any target with false discovery rate (FDR) < 10% in either direction as significant. We observed 772 significantly differentially active targets: 546 upregulated targets and 226 downregulated targets (*Figure 2A*, *Supplementary file 1*), showing that caffeine overall increases the activity of the regulatory elements.

We then focused on differentially active targets containing a known caffeine response factor as determined based on chromatin accessibility data from endothelial cells treated with caffeine (*Findley et al., 2019*). We observed that these targets had lower p-values, as observed in the QQ plot in *Figure 2A* (inset). To identify any additional transcription factors that may be important for the response to caffeine, we conducted a motif scan for 838 known transcription factor binding motifs using the JASPAR CORE Vertebrates 2022 database (*Castro-Mondragon et al., 2022*; *Supplementary file 2*). We found 19 motifs that were enriched for being within differentially active targets (*Figure 2B*, *Supplementary file 3*). We found the motif for ZNF423, one of the caffeine response factors, was enriched within the differentially active targets. The three most enriched motifs were NFATC1, NFATC4, and NFATC2. The NFAT transcription factor family is known for their involvement in the $Ca^{2+}$/NFAT pathway. This signaling pathway plays an important role in maintaining the homeostasis of vascular endothelial cells (*Wang et al., 2020*) and contributes to the mediation of proliferation and migration (*Johnson et al., 2003*; *Wang et al., 2020*). Thus, improper signaling of the $Ca^{2+}$/NFAT pathway can induce endothelial dysfunction (*Garcia-Vaz et al., 2020*; *Wang et al., 2020*). In diabetic mice, NFAT expression exacerbated atherosclerosis (*Blanco et al., 2018*; *Zetterqvist et al., 2014*) and increased foam cell formation (*Du et al., 2021*). In human coronary artery smooth muscle cells, NFAT signaling mediates vascular calcification (*Goettsch et al., 2011*). To better understand the regulatory response to caffeine, we then investigated which motifs were enriched for being within upregulated or downregulated targets separately (*Supplementary file 3*). We observed 19 motifs enriched for being within upregulated targets (*Figure 2C*) and 23 motifs enriched for being within downregulated targets (*Figure 2D*). Motifs enriched for being within upregulated targets include the previously mentioned NFATC1 and ZNF423. Motifs enriched for being within downregulated targets include the previously mentioned NFATC2 and ZNF423. We also observed that the motif for SREBF2, also called SREBP2, is enriched for being within downregulated targets. In hepatocytes, caffeine is known to suppress SREBF2 activity, which reduces PCSK9 expression, and thus increases LDLR expression, which could be protective against CVD (*Lebeau et al., 2022*). The corresponding transcription factors for these motifs could also play a role in mediating the response to caffeine in vascular endothelial cells. The motif for TEAD4 was also identified as enriched for being within downregulated targets. Interestingly, a CAD GWAS risk variant disrupts binding of TEAD4 in smooth muscle cells, causing lower expression of p16, which could potentially contribute to the risk identified at this locus (*Almontashiri et al., 2015*).

## Allelic effects on gene regulation within conditions and in response to caffeine

To investigate how genetic variation affects regulatory sequences and their function in cells treated with caffeine and in the control samples, we tested for ASE. Since the library contained the same sequence in both the forward and reverse orientations in independent targets and the regulatory effect may be direction-dependent, we tested for ASE in each SNP/direction pair separately (*Supplementary file 4*; *Figure 3—figure supplement 1*, see 'BiT-STARR-Seq Library Design' section for a more detailed description of terminology used). We observed 689 SNP/direction pairs (corresponding to 665 SNPs) with significant ASE out of 50,914 SNP/direction pairs (30,680 SNPs) tested (2.2%, FDR < 10%) (*Figure 3A*). Additionally, our library contained negative control sequences that were predicted to not have an allelic regulatory effect. These negative control sequences tend to have higher p-values than other sequences in our library, as predicted (*Figure 3A*). These results demonstrate that genetic variation within regulatory sequences within our library can modulate gene expression levels in vascular endothelial cells.

To directly test for GxE in the molecular response to caffeine, we tested for conditional allele-specific effects (cASE), where ASE is only significant in one condition, or significantly different between the two conditions. When testing for cASE, we observed 6 significant SNP/direction pairs (corresponding to 6

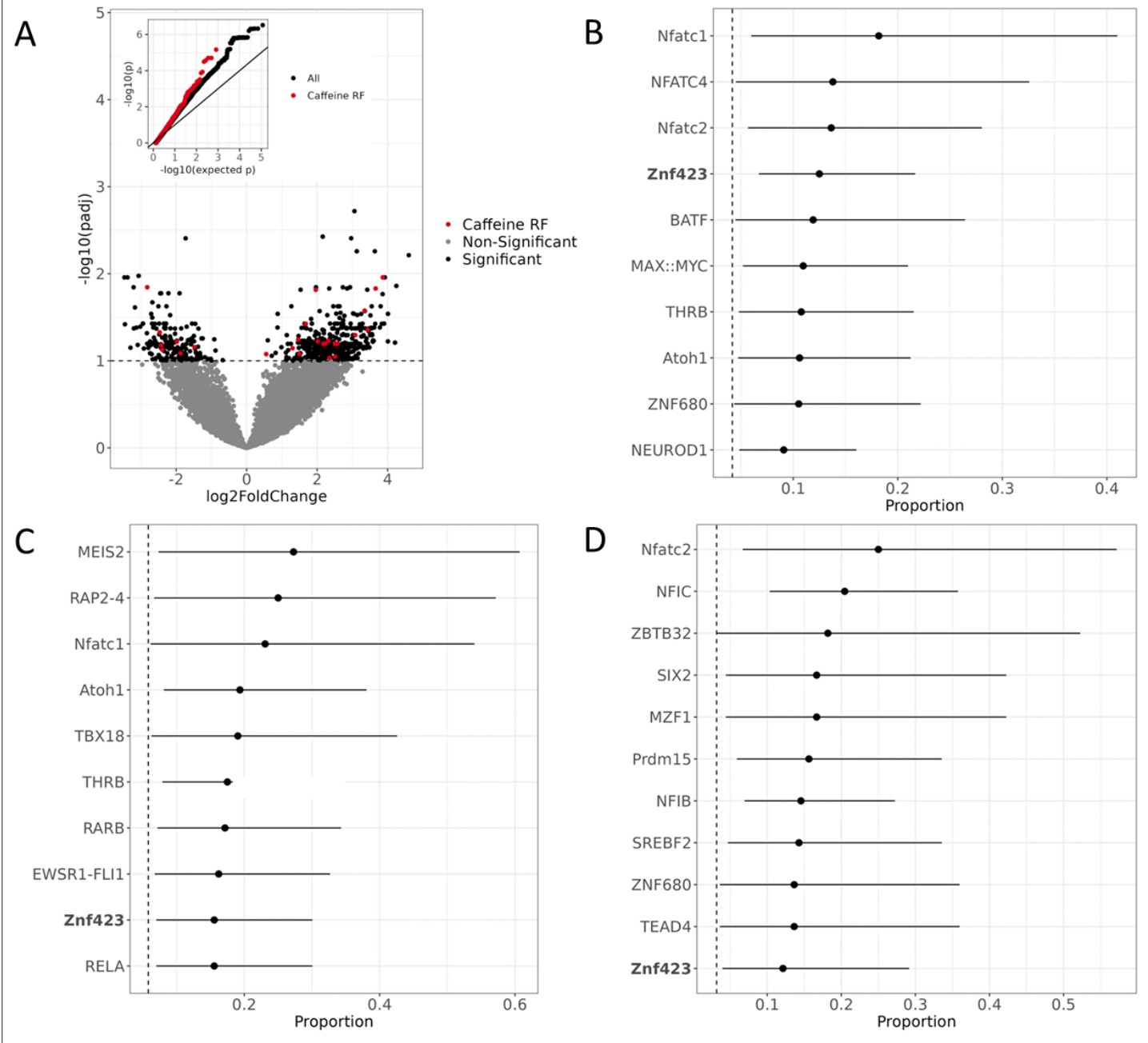

**Figure 2.** Active regulatory regions in caffeine response in vascular endothelial cells. (**A**) Volcano plot for DESeq2 results showing targets differentially active in caffeine. The light red points are significant (false discovery rate [FDR] < 10%) targets containing caffeine response factor binding sites, the black points are significant targets not containing a caffeine response factor binding site, and the gray points are nonsignificant targets. The inset contains a QQ plot for targets containing a caffeine response factor binding site (red), or no caffeine response factor binding site (black). (**B**) Motifs enriched via test of proportions (p<0.05) within differentially active targets. Names of caffeine response factors are bolded. For B-D panels error bars represent the 95% confidence interval (motif occurrence, n ≥100). (**C**) Motifs enriched via test of proportions within upregulated targets (p<0.05). (**D**) Motifs enriched via test of proportions within downregulated targets (p<0.05).

The online version of this article includes the following figure supplement(s) for figure 2:

**Figure supplement 1.** Principal component analysis (PCA) plots from read count data in the first (**A**) and second (**B**) batch (experiment), annotated by direction.

**Figure supplement 2.** Principal component analysis (PCA) plots from read count data in the first (**A**) and second (**B**) batch (experiment), annotated by allele.

**Figure supplement 3.** Heatmap plots from read count data in the first (**A**) and second (**B**) batch (experiment).

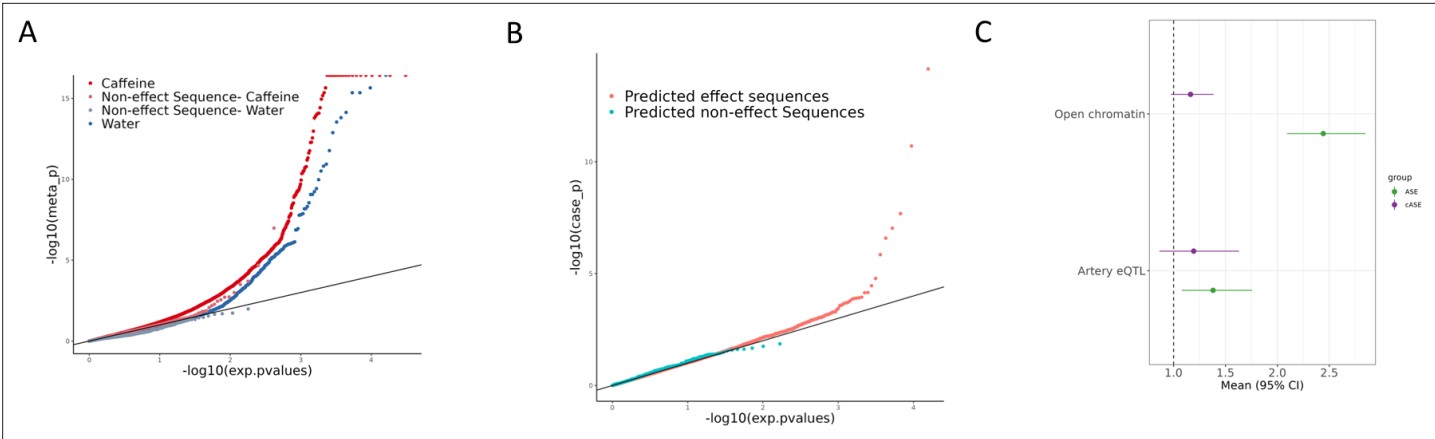

**Figure 3.** Allelic effects on gene regulation within conditions and in response to caffeine. (**A**) QQ plot depicting the p-values for allele-specific effects (ASE), with negative control sequences labeled in lighter red (caffeine) or lighter blue (control). (**B**) QQ plot depicting the p-values for conditional allele-specific effects (cASE), with targets containing caffeine response factor motifs annotated in pink and all other sequences in blue. (**C**) Enrichment via Fisher's exact test (p<0.05) of ASE (green) and cASE (purple) variants in open chromatin regions and targets containing artery expression quantitative trait loci (eQTL). Error bars indicate 95% confidence intervals. The sample sizes for each test are reported in *Supplementary file 3*.

The online version of this article includes the following figure supplement(s) for figure 3:

**Figure supplement 1.** Z-score scatter plot from allele-specific effect (ASE) analysis comparing caffeine and water.

**Figure supplement 2.** Distribution of $Z_T$-$Z_C$ (case_z) scores from conditional allele-specific effect (cASE) analysis.

**Figure supplement 3.** Artery expression quantitative trait loci (eQTL) enrichment via Torus for footprint single-nucleotide polymorphisms (SNPs) (control, black) or caffeine response factors (caffeine, red).

SNPs) out of 23,814 (15,927 SNPs) tested (FDR < 5%) (*Figure 3B*, *Supplementary file 4*, *Figure 3— figure supplement 2*). Additionally, we annotated which of these cASE targets contains a known caffeine response factor as defined in *Findley et al., 2019*, based on ATAC-seq data. There is an enrichment for these targets containing caffeine response factors, as expected (*Figure 3B*). The size of this enrichment may be underestimated due to the relatively small amount of caffeine response factor motifs (4) present in the designed library. Importantly, these variants contribute to inter-individual variation in response to caffeine. Thus, we sought to further characterize these variants.

To investigate the regulatory architecture underlying these genetic effects on gene expression, we asked whether ASE (FDR < 10%) and cASE variants (here defined at a nominal p<0.0215, N = 569) were enriched in open chromatin regions as annotated in *Findley et al., 2019*. ASE variants were 2.4-fold enriched within open chromatin regions (p<2.2e-16, *Figure 3C*, green), while a more moderate trend was observed for cASE (1.2-fold, p=0.088, *Figure 3C*, purple). This difference in enrichment could be due to the difference between the native chromatin context versus the reporter assay context. Environmental effects on gene regulatory sequences may have a more complex regulatory architecture influenced by the chromatin context that may explain the difference in the enrichment results between ASE and cASE (*Supplementary file 3*).

Genetic regulation of gene expression can be context-dependent, with factors such as cell type (*Donovan et al., 2020*; *Kim-Hellmuth et al., 2020*), developmental states (*Cuomo et al., 2020*; *Strober et al., 2019*), and environmental stimuli all contributing to GxE (GxE-eQTLs, also known as response eQTLs, dynamic eQTLs, context-eQTLs) (e.g., see *Alasoo et al., 2019*; *Barreiro et al., 2012*; *Çalışkan et al., 2015*; *Findley et al., 2021*; *Kim-Hellmuth et al., 2017*; *Maranville et al., 2011*; *Moyerbrailean et al., 2016b*). These context-specific effects can be captured without large cohorts if the appropriate experimental design is applied (*Findley et al., 2021*). Allele-specific expression experiments in two different conditions can detect GxE in small sample sizes compared to eQTL studies (*Moyerbrailean et al., 2016b*). To investigate the abundance of GxE missing from large databases such as GTEx (*Consortium, 2020*), we tested if cASE variants were enriched for artery eQTLs. Using variants within the open chromatin regions described above, we conducted a Fisher's exact test and found that ASE variants were 1.38 times more likely to be artery eQTLs (p=0.01, *Figure 3C*,

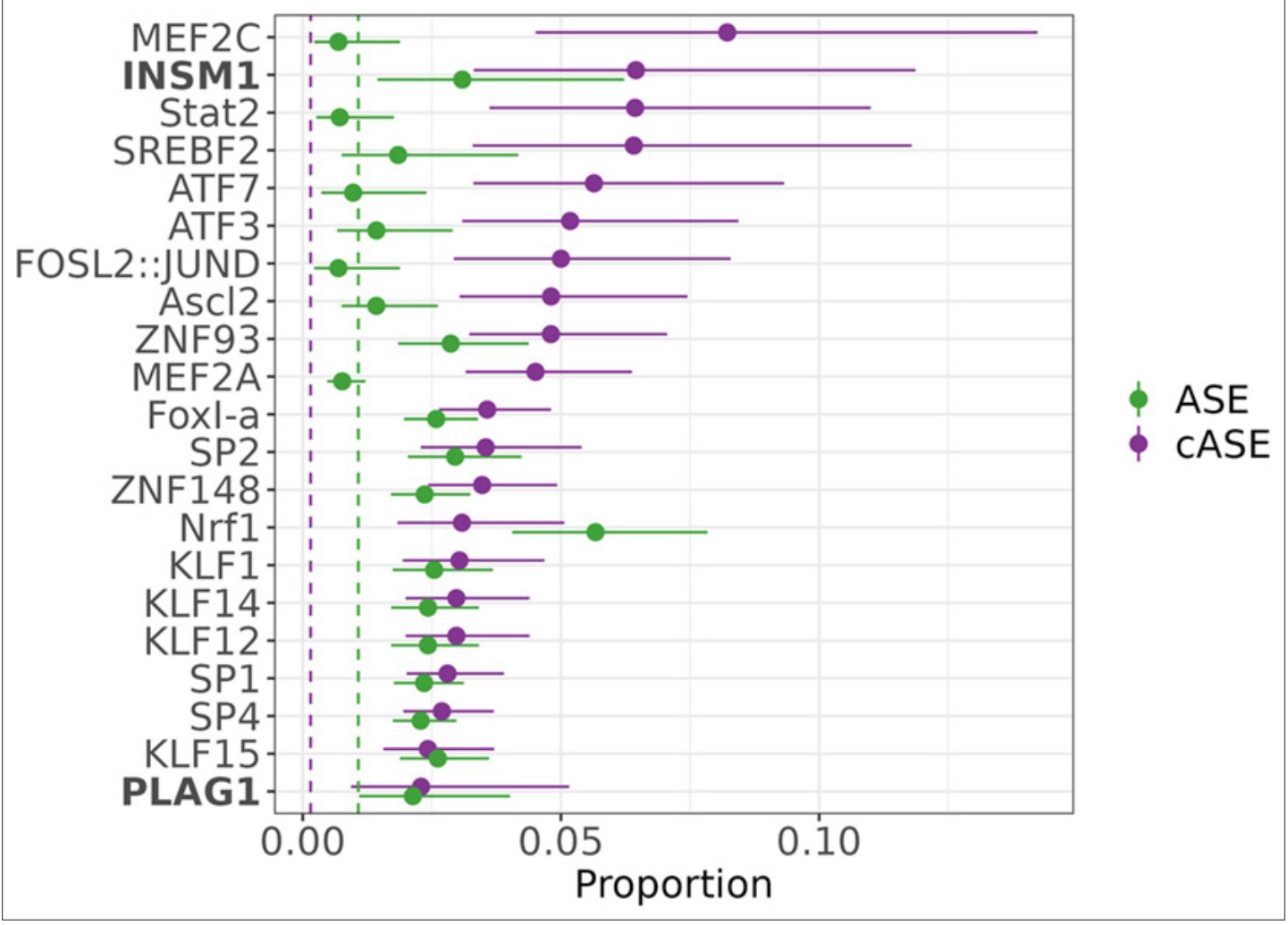

**Figure 4.** Transcription factors contributing to allele-specific effect (ASE) and conditional allele-specific effect (cASE). Motifs enriched via test of proportions (p<0.05) for significant ASE (green) or cASE (purple). The dotted lines represent the baseline proportion (mean number of significant variants within any motif) for ASE (green) and cASE (purple). Bolded factors are caffeine response factors as defined in *Findley et al., 2019*. Error bars represent the 95% confidence interval, motif occurrence n≥100.

green) compared to variants that did not show ASE. We observed a more moderate trend for cASE variants (1.2 times more likely to be artery eQTLs, p=0.26, *Figure 3C*, purple, *Supplementary file 3*).

## Characterizing ASE and cASE across transcription factor motifs

We hypothesized that the regulatory context defined by the transcription factor motifs present in each target determines the effect of a genetic variant on expression in caffeine treated cells. We conducted a motif scan of the library of targets for 838 known transcription factor binding motifs from JASPAR (*Castro-Mondragon et al., 2022*; *Supplementary file 2*). We then used a test of proportions to identify any motifs that were disproportionately within targets containing significant ASE or cASE variants. For targets containing ASE variants, we observed 44 enriched motifs (*Figure 4*). For targets containing cASE variants, we observed 18 enriched motifs (*Figure 4*, *Supplementary file 3* ). Factors of interest for cardiovascular function include NRF1, enriched for targets containing cASE and ASE variants, which is known to regulate lipid metabolism (*Hirotsu et al., 2012*; *Huss and Kelly, 2004*), and is annotated as part of the lipid metabolism pathway in Reactome (*Fabregat et al., 2018*). KLF15 and KLF14 are also enriched in targets with cASE and ASE. KLF15 is involved in cardiac lipid metabolism (*Prosdocimo et al., 2014*; *Prosdocimo et al., 2015*), and KLF14 has previously been associated with CVD (*Fryar et al., 2012*; *Hu et al., 2018*). Lastly, SREBF2, which was identified as enriched within our

differential activity results, also has implications for disease as discussed previously. This implies that SREBF2 are important both in the interindividual response to caffeine as well as disease state, linking our identified GxE with atherosclerotic disease.

We also wanted to investigate if ASE and cASE variants were disproportionately present in caffeine response factor binding sites, which may indicate that caffeine response factors' regulatory function may be modified by genetic variation. For this analysis, we used annotations from *Findley et al., 2019*, which defines caffeine response factors as transcription factors with motifs that were significantly enriched or depleted in differentially accessible chromatin after treatment with caffeine. Factors INSM1 and PLAG1 are caffeine response factors as defined in *Findley et al., 2019*, confirming that genetic variation may modulate the response to caffeine by increasing binding activity of these transcription factors.

## Validation of the regulatory mechanism for fine-mapped artery eQTLs

Computational fine-mapping is a commonly used method to identify causal variants, often for complex traits; however, further functional validation is usually needed to confirm the regulatory mechanism underlying fine-mapped causal variants. We previously showed that artery eQTLs are enriched in caffeine response factor motifs (also see *Figure 3—figure supplement 3*; *Findley et al., 2019*). We now leverage this finding to fine-map artery eQTLs using DAP-G and the caffeine response factor annotation from *Findley et al., 2019*. In our library, we tested 187 fine-mapped variants. We identified significant ASE for six SNPs (six SNP/direction pairs), thus validating the regulatory function of these fine-mapped causal eQTLs (*Supplementary file 5*). We also identified two fine-mapped artery eQTLs with significant cASE (p<0.0215; 2 SNP/direction; *Supplementary file 5*), which may represent hidden GxE in GTEx.

We then investigated if the genes shown to be linked to CAD and hypertension risk using both TWAS and colocalization analysis (INTACT, *Okamoto et al., 2023*) can be further modulated by GxE

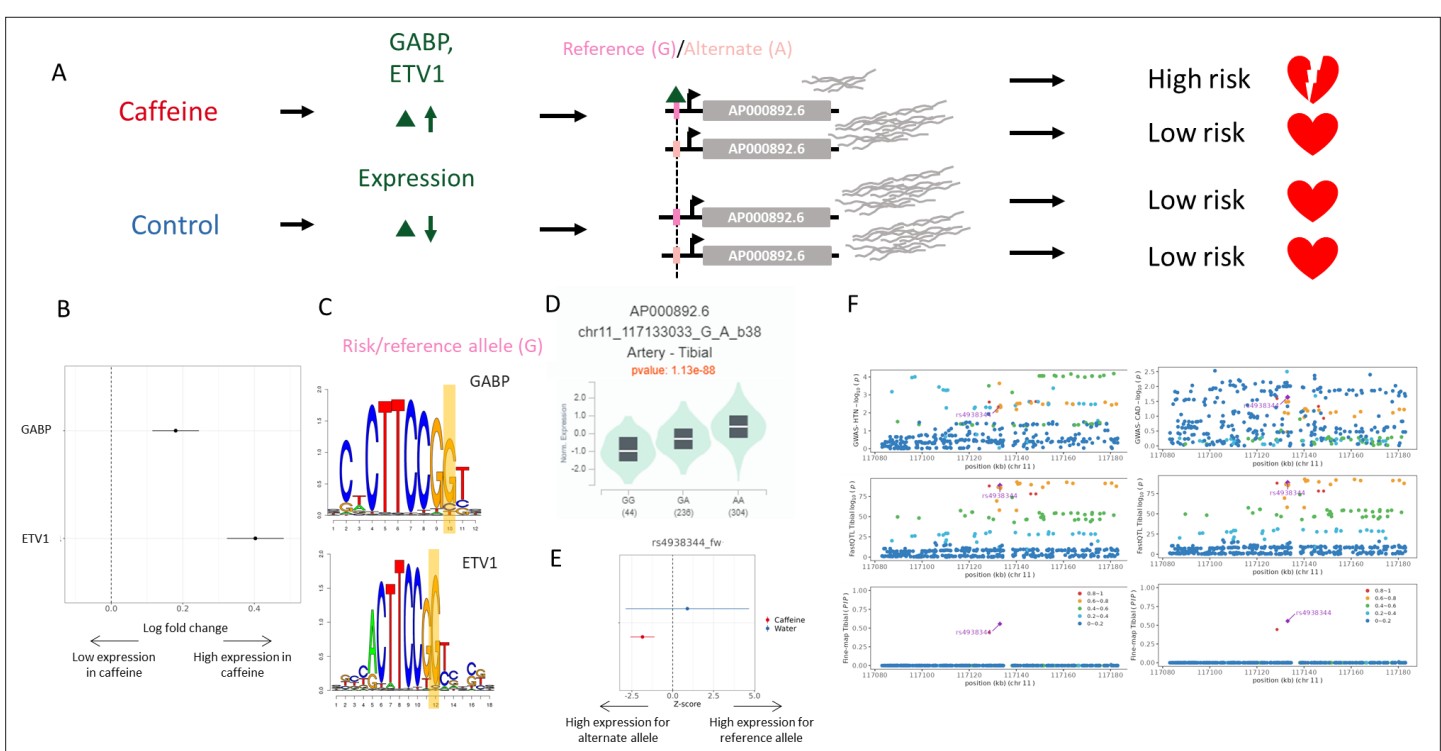

**Figure 5.** Example genetic variant with GxE with caffeine where caffeine may increase genetic risk of CAD. (**A**) Potential mechanism for rs4938344. (**B**) Transcription factors containing rs4938344 in a binding site are upregulated (via DESeq2) upon caffeine exposure (error bars are +/- standard error, FDR<10%, n=14). (**C**) Logos of transcription factor motifs with rs4938344 highlighted. (**D**) GTEx violin plot for AP000892.6. (**E**) Effect size from the BiT-STARR-Seq assay for this single-nucleotide polymorphism (SNP) within each condition (error bars are +/- standard error, n=4 replicates per condition, cASE p<0.0215, see 'cASE Analysis' section of methods). (**F**) Locus zoom plots showing genome-wide association studies (GWAS) and expression quantitative trait loci (eQTL) data for hypertension (left) and coronary artery disease (CAD) (right) in tibial artery tissue.

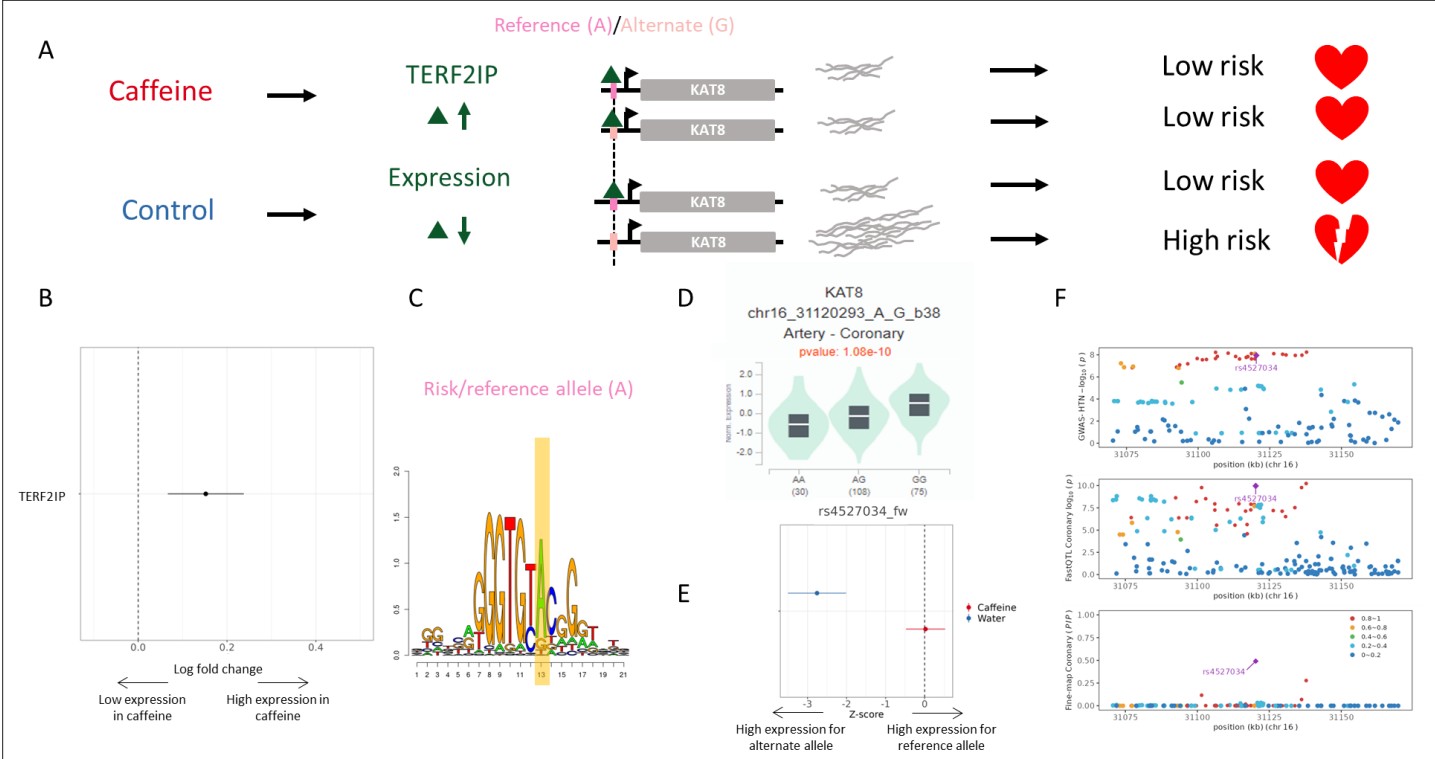

**Figure 6.** Example genetic variant with GxE with caffeine where caffeine may decrease genetic risk of CAD. (**A**) Potential mechanism for rs4527034. (**B**) TERF2IP is upregulated (via DESeq2) upon caffeine exposure (error bars are +/- standard error, FDR<10%, n = 14). (**C**) Logos of TERF2IP motif with rs4527034 highlighted. (**D**) GTEx violin plot for *KAT8*. (**E**) Effect size from the BiT-STARR-Seq assay for this single-nucleotide polymorphism (SNP) within each condition (error bars are +/- standard error, n=4 replicates per condition, cASE p<0.0215, see 'cASE Analysis' section of methods). (**F**) Locus zoom plots showing genome-wide association studies (GWAS) and expression quantitative trait loci (eQTL) data for hypertension in coronary artery tissue.

with caffeine. INTACT combines TWAS and colocalization approaches, thus overcoming the TWAS inherent linkage disequilibrium limitations. We identified eight cASE variants that regulate the expression of genes associated with CAD and hypertension (<10% FDR for INTACT analysis, p<0.0215 for cASE analysis, ***Supplementary file 6***). We used available experimental data and computational predictions to dissect the regulatory mechanisms underlying context-specific allelic effects for two of these SNPs and their impact on CAD risk (***Figures 5 and 6***). We considered allelic effects on transcription factor binding, expression of the transcription factors, and disease risk. Specifically, we considered (i) predicted effects on transcription factor binding from CentiSNP, which uses a combination of sequence information and chromatin accessibility data to predict alleles that increase binding of specific transcription factors (***Moyerbrailean et al., 2016a***); (ii) changes in the expression of the genes encoding for the relevant transcription factors and changes in chromatin accessibility (***Findley et al., 2019***); and (iii) allelic effect on the target genes (this study and GTEx).

rs4938344 is an eQTL regulating the long noncoding RNA *AP000892.6*. The reference allele at this locus, G, results in decreased expression of *AP000892.6* (as measured in GTEx and in the caffeine condition of our assay, ***Figure 5D and E***, respectively). INTACT associated high expression of *AP000892.6* with decreased risk of hypertension and CAD (***Figure 5F***). This SNP is predicted to modulate binding of GABP (a known repressor of transcription; ***Genuario and Perry, 1996***) and ETV1 at this site (***Figure 5C***). These transcription factors are upregulated in caffeine-exposed endothelial cells (***Findley et al., 2019***; ***Figure 5B***). This increase in expression uncovers allelic differences in gene regulation which are not detected in the absence of caffeine, likely because of the low expression of the repressor. The allelic differences in binding of these factors should lead to allelic differences in the expression of *AP000892.6*. Accordingly, the reference allele for this variant exhibited lower activity in response to caffeine in our BiT-STARR-Seq experiments (***Figure 5E***). This effect is consistent with the GTEx artery eQTL for *AP000892.6* (***Figure 5D***). In summary, caffeine induces higher expression of the ETV1 and GABP transcription factors, which then bind preferentially to the reference allele at

rs4938344; this results in lower expression of *AP000892.6* and increased risk for CAD and hypertension (*Figure 5A*). *AP000892.6* interacts with the *RB1CC1* RNA (*Gong et al., 2018*), which may play a role in atherosclerosis via its function in forming the autophagosome (*Chen et al., 2021*).

rs4527034 is an eQTL regulating the *KAT8* gene. The reference allele at this locus, A, results in decreased expression of *KAT8* (as measured in GTEx and in the control condition of our assay, *Figure 6D and E*, respectively). INTACT associated high expression of *KAT8* with increased risk of hypertension (*Figure 6F*). This SNP is predicted to modulate binding of the TERF2IP transcription factor at this site (*Figure 6C*). TERF2IP is upregulated in caffeine-exposed endothelial cells (*Findley et al., 2019*; *Figure 6B*). This increase in expression may saturate all binding sites in the caffeine condition, while the transcription factor may only bind to the preferential allele in the control condition. The allelic differences in binding of these factors should lead to allelic differences in expression of *KAT8* in the control condition, which is what we observe both in our BiT-STARR-Seq experiments (*Figure 6E*) and in GTEx artery eQTL for *KAT8* (*Figure 6D*). In summary, in the absence of caffeine, TERF2IP binds preferentially to the reference allele at rs4527034, which results in lower expression of *KAT8* and reduced risk for hypertension. In the presence of caffeine, TERF2IP is upregulated, resulting in increased binding and lower expression of KAT8, independently of the genotype, with an expected overall protective effect on hypertension. Confirming this potential mechanism for disease risk, TERF2IP expression levels were found to affect plaque formation in a mouse model (*Kotla et al., 2019*). High expression of KAT8, a histone acetyltransferase, also coincides with atherosclerotic progression, and histone acetylation increases in plaques within vascular endothelial cells (*Greißel et al., 2016*; *Zhang et al., 2018*).

## Discussion

This study utilized the MPRA BiT-STARR-Seq to identify gene regulatory activity in vascular endothelial cells exposed to caffeine. By utilizing BiT-STARR-Seq, we were able to identify a molecular response to caffeine, ASE, and cASE. By combining our results with preexisting annotations, we were able to characterize variants exhibiting cASE and identify potential mechanisms for some of these variants.

Heart disease is one of the leading causes of death in the United States according to the CDC (http://wonder.cdc.gov/ucd-icd10.html). The most common type of heart disease is CAD, which affects over 18 million adults over the age of 20 y (*Fryar et al., 2012*). The common risk factors of CAD include hypertension, high cholesterol levels, and family history (*Brown et al., 2022*; *Hajar, 2017*). CAD occurs when plaques form in the arteries (atherosclerosis), causing a narrowing of the artery, which reduces blood flow to the heart. The innermost layer of the artery is composed of endothelial cells. The endothelium is involved in several important functions, including regulation of blood flow, angiogenesis, thrombosis, and coagulation (*Hadi et al., 2005*; *Krüger-Genge et al., 2019*). Endothelial dysfunction occurs in diseases such as atherosclerosis and hypertension (*Xu et al., 2021*), eventually leading to CAD (*Matsuzawa and Lerman, 2014*). The molecular mechanisms behind endothelial dysfunction and the resulting diseases are largely unknown. Characterizing these molecular mechanisms is crucial in order to gain a more complete understanding of these disease phenotypes. Additionally, although caffeine is known to produce a vascular response, the current literature does not come to a consensus on the role of caffeine in CAD risk. Here, we characterized the regulatory response of noncoding variants to caffeine in vascular endothelial cells using an MPRA.

BiT-STARR-Seq, the MPRA used in this study, has several advantages over other methods used to detect GxE. One common method of detecting GxE is response eQTL mapping, which includes collecting samples from large cohorts and exposing those cells to environmental perturbations (e.g., see *Alasoo et al., 2019*; *Alasoo et al., 2018*; *Barreiro et al., 2012*; *Çalışkan et al., 2015*; *Fairfax et al., 2014*; *Huang et al., 2020*; *Kim-Hellmuth et al., 2017*; *Knowles et al., 2018*; *Lee et al., 2014*; *Mangravite et al., 2013*; *Manry et al., 2017*; *Maranville et al., 2011*; *Nédélec et al., 2016*; *Quach et al., 2016*). This method has several disadvantages as it cannot easily interrogate rare variants, relies on variation existing in a cohort (instead of investigating variants of interest), and requires larger cohort sizes to have enough power to detect GxE. In contrast, because our method uses a designed library of targets (*Kalita et al., 2018*), we are able to interrogate rare variants easily as our targets are synthesized. Similarly, we can design a library of specific variants to investigate (such as candidate regulatory variants) instead of relying on variation within a cohort (*Kalita et al., 2018*). BiT-STARR-Seq also allows us to directly compare two alleles within the same sequence context without requiring a

large cohort. Despite the advantages of BiT-STARR-Seq, unlike response eQTL mapping, we are not interrogating these SNPs in their native chromatin context. Future work may include using genome-editing tools such as CRISPR to directly insert the desired variants in their endogenous locations in the genome. For our study, we determined BiT-STARR-Seq to be the ideal assay to determine GxE for a large number of SNPs.

We observed a regulatory response to caffeine treatment, consistent with previous studies in the same cell type (*Findley et al., 2019*; *Moyerbrailean et al., 2016b*). These results suggest that caffeine exposure significantly changes the regulatory activity of vascular endothelial cells, which may have important implications regarding the impact of lifestyle in CAD. As caffeine may modulate gene regulatory activity, the resulting impact on gene expression may increase or decrease CAD risk. In addition, we identified novel transcription factors contributing to the regulatory response to caffeine including several NFAT transcription factors, and SREBF2. NFAT transcription factors are largely known for their role in the $Ca^{2+}$/NFAT signaling pathway, where $Ca^{2+}$ binds to calmodulin, stimulating calcineurin, which then causes NFAT factors to localize in the nucleus (*Crabtree and Olson, 2002*; *Klee et al., 1998*). Caffeine is known to cause an increase in $Ca^{2+}$ in human aortic endothelial cells (*Corda et al., 1995*), so it is understandable that we find these factors enriched for being within targets that respond to caffeine exposure. SREBF2, also known as SREBP2, is involved in sterol homeostasis (*Horton et al., 2003*). This result implies these novel transcription factors important for the regulatory response to caffeine may also contribute to understanding the role of caffeine in CAD risk. This coincides with findings that caffeine exposure can alter expression of genes, including those for transcription factors in mouse cardiomyocytes (*Fang et al., 2014*). Another study aimed to uncover mechanisms relevant to CVD upon caffeine exposure and found that caffeine inhibits the transcription factor SREBP2, which causes an overall protective effect against CVD (*Lebeau et al., 2022*). These results coincide with our findings.

Noncoding regions of the genome contain regulatory variants that modulate gene expression. In this study, we identify and characterize over 600 variants exhibiting ASE. Numerous noncoding variants have been implicated in CAD risk via GWAS (*Hartmann et al., 2022*; *Kessler and Schunkert, 2021*; *Koyama et al., 2020*; *Nikpay et al., 2015*; *Temprano-Sagrera et al., 2022*; *van der Harst and Verweij, 2018*), but they are generally uncharacterized. Few lead noncoding variants have been thoroughly investigated. One of these led the authors to propose and validate a molecular mechanism connecting expression of the EDN1 gene to the phenotypic outcome (*Gupta et al., 2017*; *Wang and Musunuru, 2018*). The specific mechanisms that detail how noncoding variants contribute to CAD will be critical in understanding CAD risk and ultimately developing clinical treatments. While understanding various genetic risk factors for CAD is important, GxE for these variants also have an impact on phenotype and have not been widely studied.

Since previous studies have shown that GxE-eQTL can modulate complex disease risk, we expect that GxE detected in our assay may be relevant to CAD (*Alasoo et al., 2019*; *Alasoo et al., 2018*; *Barreiro et al., 2012*; *Çalışkan et al., 2015*; *Fairfax et al., 2014*; *Findley et al., 2021*; *Huang et al., 2020*; *Kim-Hellmuth et al., 2017*; *Knowles et al., 2018*; *Lee et al., 2014*; *Mangravite et al., 2013*; *Manry et al., 2017*; *Maranville et al., 2011*; *Nédélec et al., 2016*; *Quach et al., 2016*). We tested for cASE, which occurs when ASE are only significant in one condition, or significantly different between the two conditions. This analysis identifies GxE which are important in understanding disease risk while accounting for genetic and environmental context. We identified 6 variants that regulate the gene expression response to caffeine and demonstrated that context-aware MPRAs can be used to dissect molecular mechanisms underlying cardiovascular health.

By fine-mapping artery eQTLs and combining the data collected from our assay with preexisting annotations, we produced potential mechanisms for two cASE variants through altered transcription factor expression and binding in response to caffeine. Our results indicated that both genetic and environmental factors are important in determining risk, and that the interaction between these factors can be informative to mechanisms and phenotypic consequences. Importantly, by utilizing multiple functional annotations, we are able to identify variants that may be relevant to disease but did not reach genome-wide significance in GWAS, possibly because of their context-specific effects. By studying different environmental contexts, we can identify that, in these instances, the presence of caffeine can impact the risk of poor cardiovascular health outcomes. If environmental context was not considered and this work was conducted solely in the control condition, the caffeine modulatory effect on risk would have been missed.

Although we investigate GxE for caffeine in vascular endothelial cells, our experimental approach can be applied to various different complex diseases and their relevant cell types and treatments. To further validate our work, genome-editing tools could be used to investigate the effect of these variants in their native chromatin context. Additional validation could include allele-specific and condition-specific transcription factor binding assays (such as electrophoretic mobility shift assays) for the fine-mapped variants.

Our study demonstrates the importance of considering environmental contexts when investigating gene regulatory activity as we identify several thousand instances of GxE in our library of candidate regulatory variants. Our data, combined with preexisting annotations, allowed us to identify transcription factors involved in GxE in caffeine and describe specific potential molecular mechanisms for some of these GxE. Our results provide important insights into the molecular regulatory effect of caffeine exposure and GxE for caffeine in vascular endothelial cells.

## Materials and methods

### Cell culture

Human umbilical vein endothelial cells (HUVECs) were obtained from Lonza (Cat# CC-2517-0000315288). Cells were cultured at 37°C with 5% $CO_2$ and seeded at 5000 cells/cm$^2$. EGM-2 growth medium (Lonza) was used to culture the cells.

### Treatment

Treatment concentration was the same as used in previous studies (*Findley et al., 2019*; *Moyerbrailean et al., 2016b*). We used a caffeine concentration of $1.16 \times 10^{-3}$ M. In addition, water was used as a vehicle control as that was the solvent used to prepare the caffeine treatment.

### BiT-STARR-Seq library design

We designed 43,556 target regulatory regions each containing an SNP in the middle and with a total length of 200 nucleotides. This set of targets corresponds to 87,112 constructs each containing only one of two alleles at the test SNP. Additionally, each construct can be integrated in the forward or reverse orientation, leading to a maximum of 174,224 constructs in either direction. Please also see below for a description of how we use library-related terms throughout the article. The library used is the same as reported in *Kalita et al., 2018*. Briefly, the library of target regulatory sequences consisted of several categories of regulatory variants, including eQTLs (*Innocenti et al., 2011*; *Wen et al., 2015*), SNPs predicted to disrupt transcription factor binding (centiSNPs) (*Moyerbrailean et al., 2016a*), and SNPs associated with complex traits in GWAS (*Pickrell, 2014*). Negative controls that were not predicted to have a regulatory effect were also included in the library (*Moyerbrailean et al., 2016a*). It is important to note that these negative controls are only predicted not to have a regulatory effect via computational annotation (*Moyerbrailean et al., 2016a*), so they may not be representative of true negative controls. This is why we largely do not utilize these SNPs as negative controls within our analyses. Our predictions of regulatory activity also did not account for environmental context, thus these sequences are also not suited to annotate our cASE results.

> SNP (n = 43,556): Refers to a genetic variant tested for allelic effects on gene regulation.
> Target (n = 43,556): 200-nucleotide-long oligonucleotide sequence that contains the test SNP in the middle of the target.
> Construct (n = 87,112): Synthesized 200-nucleotide-long oligonucleotide sequence that contains only one of the two possible alleles at the test SNP. Each target corresponds to two constructs.
> Direction: Constructs can integrate in either the forward or reverse direction relative to the direction of transcription in the BiT-STARR-Seq plasmid. Therefore, two directions are possible for each construct.
> SNP/direction pair (n = 87,112): An SNP tested for allelic effects on gene regulation contrasting the expression of two constructs that are integrated in the same direction. All statistical tests are performed at this level, testing in each direction separately.

## BiT-STARR-Seq experiments

Oligonucleotides were synthesized and used to create a DNA plasmid library, which was sequenced and used as a subsequent input for the ASE analysis. The DNA library was transfected into HUVECs using the Lonza Nucleofector X platform. Cells were electroporated using the DS-120 setting with primary cell solution P5. Caffeine was added at $1.16 \times 10^{-3}$ M after transfection. Cells were incubated for 24 hr and lysed. We completed 6 replicates per treatment condition (caffeine and the water vehicle control) or 12 replicates in total.

## Library preparation and sequencing

RNA was extracted using the RNeasy Plus Mini kit (QIAGEN, Cat# 74136). A cDNA library was prepared using the Superscript III First-Strand Synthesis kit (Invitrogen, Cat# 18080-400). Sequencing was completed using the Illumina Nextseq 500 to generate 125 cycles for read 1, 30 cycles for read 2, 8 cycles for the fixed multiplexing index 2, and 10 cycles for index 1 (variable barcode). The average sequencing depth per library was 39,235,611 reads, for a total of 470,827,333 reads (*Figure 1—figure supplement 1*, *Supplementary file 7*).

## Processing sequence data

To analyze the RNA-seq data, we began by demultiplexing our data using the bcl2fastq software to create demultiplexed FASTQ files. We then aligned to hg19 using HISAT2. Afterward, we applied a filter to ensure the UMIs present match the expected UMI pattern (RDHBVDHBVD). Reads with short UMIs or those that do not match the expected sequence were removed. The resulting BAM files were then deduplicated using UMItools. We ran samtools mpileup followed by bcftools query to output read counts per each allele/direction combination.

## Differential activity analysis

To test for a molecular response to caffeine, we used the R package DESeq2 (*Love et al., 2014*). To determine which model would best test for a molecular response to caffeine, we completed principal component analysis to identify major sources of variation. We identified that the first PC clearly represented allelic effects (*Figure 2—figure supplement 2*, also see *Figure 3—figure supplements 1 and 3*), thus we included allele (reference or alternate) as part of our model. Our model tested the effect of treatment, correcting for allele (~allele + treatment), as we observed a strong allelic effect. We ran DESeq2 for each direction (see *Figure 2—figure supplement 1*) as the library contained motifs in both the forward and reverse orientations within separate target sequences. We considered targets as significant with Benjamini–Hochberg FDR <10%.

## ASE analysis

To test for ASE, we utilized the R package quantitative allele-specific analysis of reads (QuASAR-MPRA) (*Kalita et al., 2017*). QuASAR-MPRA is an extension of the QuASAR package which allows for analysis of barcoded MPRA data. QuASAR-MPRA uses a beta-binomial model and accounts for uneven initial allelic proportions present in the DNA library. We used the fitQuasarMpra() function to test for ASE in each experiment separately, estimating the ASE effect and its standard error. For each SNP/direction pair, we meta-analyzed the effect size using a weighted mean utilizing inverse-variance weighting for each condition separately. The z-score for each SNP-direction pair is subsequently calculated as the meta-analyzed effect size minus the DNA proportion, divided by the meta-analyzed standard error of the effect size. We then required each identifier to be within four or greater replicates (out of the six total replicates) and performed multiple test correction using the Benjamini–Hochberg procedure. Significant ASE was then defined as having an FDR <10%.

## cASE analysis

To test for cASE, we used a method previously developed by our lab called differential allele-specific test, or ΔAST. The calculation for this parameter ΔZ is provided in *Moyerbrailean et al., 2016a* as well as below. The QuASAR-MPRA package outputs betas for the treatment ($\beta_T$) and the control ($\beta_C$), as well as the standard error (se) for both groups, which are used to calculate a Z score for each condition independently ($\frac{\beta}{se}$). To contrast ASE between conditions we define the cASE statistic ($\Delta Z$)$^2$, as ($\Delta Z$)$^2$ = (Z$_T$ - Z$_C$)$^2$ /2. From this ($\Delta Z$)$^2$ statistic, we used the genomic control procedure (*Devlin et al.,*

*2001*; *Devlin and Roeder, 1999*) to calibrate the median $\chi^2$ to the expected value and we calculated the p-values using the $\chi^2$ distribution. We then used the p.adjust() function to perform multiple test correction using the Benjamini–Hochberg procedure. Significant cASE was then defined as having an FDR<5%.

## Motif enrichment analysis

The motif scan was completed using the PWMScan tool, using all PWMs within the JASPAR (*Castro-Mondragon et al., 2022*) 2022 CORE database (838 motifs). A threshold of -t 10 (base 2) was used for the motif scan, which was restricted to the regions of our designed targets. Once the scan was complete, motifs that were present less than 100 times in the library were removed. For differentially active targets in response to caffeine, 222 motifs passed this filter. For motifs containing ASE variants, 359 motifs passed this filter. For motifs containing cASE variants, 417 motifs passed the filter. A test of proportion (prop.test() in R) was performed per each transcription factor, where the null proportion was the total number of significant targets/variants containing/within motifs (differentially active target, ASE or cASE) divided by the total number of nonsignificant targets/variants containing/within motifs. The test was done per motif, where the proportion being tested is the same as the null proportion, but conducted per motif rather than across all motifs. The related calculations are shown below:

Expected null proportion = $n_a/n_b$
Observed motif proportion = $n_c/n_d$
$n_a$ = number of targets/variants of interest containing/within any motif
$n_b$ = number of targets/variants containing/within any motif
$n_c$ = number of targets/variants of interest containing/within specific motif
$n_d$ = number of targets/variants containing/within specific motif

ASE (FDR <10%) and cASE (p<0.0215) were considered significant. For follow-up analyses of cASE features, we relaxed the significance threshold to nominal p-value <0.0215.

## Open chromatin region enrichment analysis

To test if certain variants were within open chromatin regions, we obtained the list of differentially accessible regions tested in *Findley et al., 2019*. We considered any accessible region (annotated as differentially accessible or not). Bedtools bed intersect tool was used to complete the overlap with the appropriate datasets (DESeq2, ASE, or cASE results), resulting in a list of targets or SNPs that were within open chromatin regions. This annotation was then used to complete the Fisher's exact test. Enrichments for significant ASE or cASE variants were performed separately.

## Artery eQTL enrichment analysis

To determine if certain variants were artery eQTLs, we obtained GTEx v8 (*Castro-Mondragon et al., 2022*) data for aorta, coronary, and tibial artery tissues. We then intersected the list of variants of interest with the list of artery eQTLs. To determine if artery eQTLs were within differentially active targets, bedtools intersect was used, resulting in a list of targets that contained artery eQTLs. We further subset this list to variants within open chromatin regions (see previous section). This annotation was then used to complete the Fisher's exact test. Enrichments for significant ASE or cASE variants were performed separately.

## Fine-mapping analysis with DAP-G

Based on a previous study (*Findley et al., 2019*), we define caffeine response factors as transcription factors with motifs that were significantly enriched or depleted in differentially accessible chromatin after treatment with caffeine. We annotated genetic variants into two categories: (1) genetic variants in motifs for response factors and (2) genetic variants in motifs for transcription factors that are not caffeine response factors. By integration of these genetic variants annotation, we estimated the probability of each SNP regulating gene expression in a Bayesian hierarchical model using TORUS (*Wen, 2016*). These probabilities are then used in DAP-G (*Zhang et al., 2020*) to fine-map eQTLs from all three artery tissues in GTEx V8. A total of 364,427,888 eQTLs were fine-mapped across three artery tissues. We filtered for a posterior inclusion probability of >0.9.

## Identification of putative causal genes

To better understand how cASE variants may impact traits related to cardiovascular health, we performed colocalization analysis of the causal GWAS variants with eQTLs using fastENLOC (*Wen et al., 2017*) by integration of fine-mapped eQTLs in the three artery tissues (see above section) and fine-mapped GWAS signals for CAD and hypertension. We fine-mapped GWAS using DAP (*Wen, 2016*). From fastENLOC, we obtained the gene locus-level colocalization probability for each gene which is used to evaluate how the gene is associated with complex traits or diseases. Intuitively, colocalization analysis identified the overlap of causal eQTLs and GWAS hits. However, it lacked the sensitivity due to the failure in distinguishing between the vertical pleiotropy (genetic effects on traits mediated by gene expression) and horizontal pleiotropy (independent effects on gene expression and traits) (*Okamoto et al., 2023*). To overcome the limitations in a single approach, we combined the evidence from colocalization and TWAS to estimate the probability of putative causal genes using the R package INTACT (*Okamoto et al., 2023*). Here, the TWAS data we utilized were from PTWAS (*Zhang et al., 2020*). We determined putative causal genes with FDR < 10%. We then identified variants regulating these putative causal genes using the DAP-G fine-mapping results, requiring a SNP-level colocalization probability >0.5, with no threshold on PIP (*Supplementary file 6*).

## Acknowledgements

We thank the members of the Luca and Pique-Regi laboratories for helpful comments and discussions.

## Additional information

### Funding

| Funder | Grant reference number | Author |
|---|---|---|
| National Institute of General Medical Sciences | R01GM109215 | Francesca Luca Roger Pique-Regi |
| National Institute of Environmental Health Sciences | R01ES033634 | Francesca Luca Roger Pique-Regi Xiaoquan Wen |

The funders had no role in study design, data collection and interpretation, or the decision to submit the work for publication.

### Author contributions

Carly Boye, Data curation, Formal analysis, Validation, Visualization, Writing – original draft, Writing – review and editing; Cynthia A Kalita, Data curation, Investigation, Methodology, Writing – review and editing; Anthony S Findley, Data curation, Formal analysis, Writing – review and editing; Adnan Alazizi, Investigation, Writing – review and editing; Julong Wei, Formal analysis, Visualization, Writing – original draft, Writing – review and editing; Xiaoquan Wen, Software, Methodology, Writing – review and editing; Roger Pique-Regi, Francesca Luca, Conceptualization, Supervision, Funding acquisition, Writing – original draft, Project administration, Writing – review and editing

### Author ORCIDs

Carly Boye http://orcid.org/0000-0002-9142-0240
Anthony S Findley http://orcid.org/0000-0001-9922-3076
Roger Pique-Regi http://orcid.org/0000-0002-1262-2275
Francesca Luca http://orcid.org/0000-0001-8252-9052

### Decision letter and Author response

Decision letter https://doi.org/10.7554/eLife.85235.sa1
Author response https://doi.org/10.7554/eLife.85235.sa2

## Additional files

### Supplementary files

• Supplementary file 1. Differential activity results. Output from DESeq2. Column 'V1' is the genomic position (hg19) of the SNP within the construct being tested, 'baseMean' refers to normalized expression, 'log2FoldChange' refers to the effect size, 'lfcSE' is the standard error of the effect size, 'stat' is the log2FoldChange divided by lfcSE, 'pvalue' is the nominal p-value, and 'padj' is the adjusted p-value. Please refer to DESeq2 documentation for additional details.

• Supplementary file 2. PWMScan results. Includes all variants in the designed library that are within a motif from the JASPAR 2022 CORE Vertebrates database determined by PWMScan. Columns 1–3 are the genomic position of the test SNP, column 4 is the rsID for the test SNP, and column 5 is the ID of the motif in JASPAR.

• Supplementary file 3. Characterization of regulatory regions and variants. Includes contingency tables for all Fisher's exact tests reported in this study. Please see the legend provided within the file. Briefly, A1–C5 is the contingency table and summary of results for ASE variants within open chromatin regions, A6–C10 for ASE variants overlapping with artery eQTLs, A11–C15 for cASE variants within open chromatin, and A16–C20 for cASE variants overlapping with artery eQTLs.

• Supplementary file 4. Allele-specific effects and conditional allele-specific effects results. Output from ASE/cASE analysis. The 'identifier' column is the SNP/direction pair, 'meta_estimate' is the effect size (of the ASE), 'meta_se' is the standard error of the effect size, 'n' is the number of replicates containing nonzero read counts, 'DNA_prop' is the proportion of reads (reference/alternate) present in the DNA library, 'meta_z' is the z-score, 'meta_p' is the nominal ASE p-value, 'meta_padj' is the adjusted ASE p-value before filtering for SNP/direction pairs where $n > 3$, 'new_padj' is the adjusted ASE p-value after filtering for SNP/direction pairs where $n > 3$, 'group' refers to the applicable condition for the statistic (.x suffix for caffeine, .y suffix for control), 'case_z' is $Z_T$-$Z_C$, 'case_p' is the nominal p-value for cASE, and 'case_padj' is the adjusted p-value for cASE.

• Supplementary file 5. Fine-mapped artery eQTLs with significant ASE or cASE. The 'chr' column contains the chromosome information, 'pos' is the genomic position (0-based, hg19), 'pos1' is the genomic position (1-based, hg19), 'identifier' is the SNP/direction pair, 'rsID' refers to rsID, 'padj_min' refers to the minimum p-adjusted value for ASE (across conditions), 'ID' is an ID to identify the genomic position in the format of chr_pos1, 'PIP' is the posterior inclusion probability as calculated by DAP-G, 'gene' refers to the Ensembl gene ID, 'tissue' refers to the GTEx artery tissues (tibial, aorta, or coronary), 'group' identifies if the variant has significant ASE or significant cASE, 'case_p' refers to the cASE nominal p-value, and 'case_padj' refers to the cASE adjusted p-value.

• Supplementary file 6. Variants that regulate putatively casual genes as identified via INTACT. The 'Identifier' column is the SNP/direction pair, 'ID' is an ID to identify the genomic position in the format of chr_pos1 (see above), 'Gene ID' refers to the Ensembl gene ID, 'Gene symbol' refers to the gene symbol, 'Trait' is CAD or hypertension (HTN), 'Tissue' refers to the GTEx artery tissues (tibial, aorta, or coronary), 'DAP-G PIP' is the posterior inclusion probability as calculated by DAP-G, 'GWAS z-score' is the z-score for the GWAS of the corresponding trait, 'GWAS pvalue' is the nominal p-value for the GWAS of the corresponding trait, 'PTWAS-INTACT PCG' is the probability of being a putative causal gene as calculated by INTACT, 'PTWAS-INTACT FDR' is the FDR statistic as calculated by INTACT, 'INTACT z-score' is the z-score statistic as calculated by INTACT, 'cASE_p' is the cASE nominal p-value, 'cASE_padj_(original)' is the cASE adjusted p-value, 'cASE_z' is $Z_T$-$Z_C$, 'DE_construct?' indicates if the construct is significantly differentially active as defined by our assay, 'Caffeine_ASE_z' is the z-score for ASE (caffeine condition), 'Water_ASE_z' is the z-score for ASE (control condition), 'centiSNP motifs' refers to motifs the SNP is within as identified by centiSNP, and 'centiSNP category' refers to if the motif is a footprintSNP, effectSNP, or switchSNP as identified by centiSNP.

• Supplementary file 7. Read counts. Read counts for all libraries at different steps of the data processing pipeline. 'Library Name' is an identifier used for the sample, 'Treatment' is the condition (caffeine or control), 'Dedup' refers to the number of reads after deduplication, and '>1 read either direction' describes the number of reads that have >1 read for each allele in either direction.

• MDAR checklist

### Data availability

FASTQ files and read count data are available at the GEO accession number GSE221514. Supplemental files are available at https://doi.org/10.5281/zenodo.7327508.

The following datasets were generated:

| Author(s) | Year | Dataset title | Dataset URL | Database and Identifier |
|---|---|---|---|---|
| Boye C, Kalita C, Findley A, Alazizi A, Wei J, Wen X, Luca F, Pique-Regi R | 2024 | Characterization of caffeine response regulatory variants in vascular endothelial cells | https://www.ncbi.nlm.nih.gov/geo/query/acc.cgi?acc=GSE221514 | NCBI Gene Expression Omnibus, GSE221514 |
| Boye C, Kalita C, Findley A, Alazizi A, Wei J, Wen X, Pique-Regi R, Luca F | 2024 | Characterization of caffeine response regulatory variants in vascular endothelial cells | https://doi.org/10.5281/zenodo.7327508 | Zenodo, 10.5281/zenodo.7327508 |

The following previously published dataset was used:

| Author(s) | Year | Dataset title | Dataset URL | Database and Identifier |
|---|---|---|---|---|
| Findley AS, Richards AL, Petrini C, Alazizi A, Shanku AG, Davis GO, Hauff N, Sorokin Y, Wen X, Luca F, Doman E, Pique-Regi R | 2019 | Gene-Environment Interactions (GxE) and Complex Traits | https://www.ncbi.nlm.nih.gov/projects/gap/cgi-bin/study.cgi?study_id=phs001176.v3.p1 | NCBI Gene Expression Omnibus, phs001176.v3.p1 |

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
