## [Editor Report]

This important study identifies context-specific regulatory variants by an MPRA screen in vascular endothelial cells exposed to caffeine. The authors use a compelling and creative approach to pinpoint potential molecular mechanisms of gene-by-environment effects on gene regulation. The variants they identify are likely linked to complex disease risk.

---

## [Decision Letter]

**Decision letter after peer review:**

Thank you for submitting your article "Characterization of caffeine response regulatory variants in vascular endothelial cells" for consideration by *eLife*. Your article has been reviewed by 3 peer reviewers, and the evaluation has been overseen by a Reviewing Editor and Detlef Weigel as the Senior Editor.

Essential revisions:

Please see the reviewer comments below, which elaborate on these overall essential revisions. Importantly, all three reviewers had a productive internal discussion and are in agreement about these revisions.

1. Clarify p-value calculations, N oligos, Figure 2 ambiguities, selection of null SNPs, and method details. Convincingly demonstrate the overall robustness of the underlying data.

2. Show a more convincing locus than the PIP4K2B example, which is only supported by a pTWAS result (with LD contamination issues) with no fine mapping or even nominal p-value evidence.

*Reviewer #1 (Recommendations for the authors):*

It would be good to have the number of ASE and cASE variants clearly stated.

Figure 1 illustrates only parts of the questions of this study. It would be helpful to include also an illustration of the regulatory element activity underwater/caffeine conditions, and allelic comparison under the two comparisons.

"Only one lead non-coding variant has been thoroughly investigated" (referring to EDN1) but what about e.g. SORT1? One could debate what's "thorough" but I'd soften this statement.

Supplemental figure 2: what is the ethanol experiment here? I'm confused.

The quality of the supplementary figures and their legends could be improved. There are some errors in references to figure panels (SFigure 2 & 3), and axis legends that are way too small (SFigure 5). Please arrange the layout so that the figure and legend are on the same page.

"Only two previous studies have used MPRAs to investigate DNA sequences that regulate the transcriptional response to treatments"- it is of course a matter of taste if preprints count here, but I'd add this one too: https://www.biorxiv.org/content/10.1101/2020.09.30.321323v1.full

*Reviewer #2 (Recommendations for the authors):*

1. After performing differential activity analysis to determine significantly differentially active constructs, the authors move to investigate the allele effects of gene regulation in both treatment conditions. one point needs clarification:

When ASE is tested for each SNP/direction in the control condition, 30096 pairs were tested, vs 49441 pairs in the caffeine treatment condition. How was the number of test pairs determined? Was this based on the number of differentially active constructs in that condition?

2. Likewise with the cASE test: how was the initial SNP list of 28922 SNPs determined?

3. In figure 5, identifying fine-mapped artery eQTLs with significant cASE, pTWAS was used to investigate the genes regulated by the eQTLs and their phenotypic consequences. Although the code is cited, it would be beneficial to have a brief summary of pTWAS, similar to the summary given for DAP-G in the methods section.

4. The legend in supplemental figure 2 says "Z-score scatter plots from ASE analysis comparing water and ethanol (A) and caffeine and water (B)". However, there doesn't appear to be a Suppl. Figure 2A and 2B, just a single scatter plot. Likewise, ethanol is mentioned here as a treatment condition and does not appear elsewhere in the manuscript.

5. The authors acknowledge the limitation of their MPRA with respect to native chromatin context in the discussion. Likewise, although HUVECs are a common model cell line for endothelial cells, they have their own limitations with respect to CAD pathophysiology. Adding this to the discussion would be beneficial.

6. In figure five the SNP rs228271 is shown as an example of the prioritization approach, along with a proposed example mechanism. It would be interesting to see if part of this mechanism can be validated. For example: if the HUVECs cells used in this study are homozygous for the alternate allele of rs228271, does caffeine treatment decrease PIP4K2B activity?

7. Columns in supplementary table 2 need to be explained.

8. There are several places where the authors cite the literature but it is not clear how the cited studies and results relate to their study. Some examples are below:

a. Authors state that "In diabetic mice, NFAT expression exacerbated atherosclerosis (Blanco et al., 2018; Zetterqvist et al., 2014) and increased foam cell formation (Du et al., 2021)" In which cells did NFAT expression exacerbated atherosclerosis? In endothelial cells or other cells? How is this study relevant to caffeine response in endothelial cells? Is there a known causal link between caffeine consumption and atherosclerosis?

b. Authors state that "In hepatocytes, caffeine is known to suppress SREBF2 activity, which reduces PCSK9 expression, and thus increases LDLR expression, which could be protective against cardiovascular disease (Lebeau et al., 2022)." How do the authors translate this knowledge to ECs and caffeine response? Does the same LDLR/PCSK9 pathway exist in ECs? Does SREBF2 regulate it? What is known about the relationship between caffeine and lipid signaling in ECs?

c. Authors state that "Factors of interest for cardiovascular function include NRF1, which is enriched for constructs containing both ASE and cASE variants, is known to regulate lipid metabolism (Hirotsu et al., 2012; Huss and Kelly, 2004), and is annotated as part of the lipid metabolism pathway in Reactome (Fabregat et al., 2018). KLF15 and KLF14 are enriched only in constructs with ASE. KLF15 is involved in cardiac lipid metabolism (Prosdocimo et al., 2014, 2015), and KLF14, has previously been associated with cardiovascular disease (Chen et al., 2012; Hu et al., 2018)." what is the relationship between lipid metabolism and caffeine response in ECs?

9. Authors perform enrichment analysis of cASE variants in GTEx artery eQTLs to ask if GTEx artery eQTLs are missing GxE eQTL effects. Artery eQTLs are enriched for gene expression from smooth muscle cells. There are endothelial eQTL datasets (PMIDs 32442411, 20170901, 23667179) Those are the appropriate datasets to test the authors' hypothesis and should be used in this study.

*Reviewer #3 (Recommendations for the authors):*

1. You include some detail about experimental characteristics including a heatmap of counts per sample and PCA plots – elaborating to explicitly describe inter-replicate correlation and range of oligo activity (not just allelic bias) would be appreciated. It would also be helpful to give even limited details about where these variants come from, including which tissues the eQTLs correspond to, which diseases/traits for GWAS, etc. to set expectations for some of the enrichment analyses that are performed.

2. Because you seem to be testing different sets of variants/construct orientations (i.e., denominators) across different analyses, it would be helpful to give greater clarity on where these numbers are coming from and to add sample sizes to your figure legends. Additionally, the abstract also points out some very specific numbers that are never mentioned in the main text (e.g., 7,152 variants that regulate the expression of caffeine, only mentioned in the discussion).

3. The methods section mentions the use of QuASAR-MPRA, which enables "analysis of barcoded MPRA data." You mention the use of UMIs to deduplicate reads (accounting for PCR duplicates), but are the oligos in this library barcoded during plasmid preparation? If so, what was the average and range of barcode depth across oligos?

4. Negative control oligos are included in the library, but no description of how they were used for the analysis is given. They could be highlighted in a different color on the heat map and/or a description could be given whether they were used for setting thresholds.

5. You show data and a proposed model for a specific variant in figure 5, but the inclusion of actual BiT-STARR-seq data would augment your interpretation and descriptive model.

6. One additional study to cite in the introduction is "Functional characterization of T2D-associated SNP effects on baseline and ER stress-responsive β cell transcriptional activation" (PMID: 34475398). This study examined treatment-specific effects on allelic bias in an MPRA, including variants known to be e- and caQTL.

7. Though you present a model for GxE at a specific variant in figure 5, the study design cartoon in figure 1 only demonstrates allelic effects. Given that context-specific effects are the focal point of the manuscript, it would help readers to illustrate some hypothetical G, E, and GxE effects on MPRA activity.

---

## [Author Response]

Essential revisions:Please see the reviewer comments below, which elaborate on these overall essential revisions. Importantly, all three reviewers had a productive internal discussion and are in agreement about these revisions.1. Clarify p-value calculations, N oligos, Figure 2 ambiguities, selection of null SNPs, and method details. Convincingly demonstrate the overall robustness of the underlying data.

We thank the reviewers and editor for these comments which we have addressed as described below:

P-value calculations: Previously, to summarize the DESeq results we had plotted the lowest pvalue per construct (either the p-value for the forward or reverse orientation). By selecting the pvalues this way, the plot seemed to have inflated p-values. Instead, we have changed the plot to show all p-values (for both directions when available).

**Author response image 1. sa2fig1:** 

N oligos and selection of null SNPs: we have clarified the nomenclature used across the manuscript to refer to SNP/direction pairs or SNPs and the number of oligos tested for each analysis. We have also clarified the selection of null SNPs. All these changes were introduced in the “BiT-STARR-Seq Library Design” section of methods and are also copied below:“We designed 43,556 target regulatory regions each containing a SNP in the middle and with a total length of 200 nucleotides. This set of targets corresponds to 87,112 constructs each containing only one of two alleles at the test SNP. Additionally, each construct can be integrated in the forward or reverse orientation, leading to a maximum of 174,224 constructs in either direction. Please also see below for a description of how we use library-related terms throughout the paper. The library used is the same as reported in Kalita et al. (Kalita et al., 2018). Briefly, the library of target regulatory sequences consisted of several categories of regulatory variants, including eQTLs (Innocenti et al., 2011; Wen et al., 2015), SNPs predicted to disrupt transcription factor binding (centiSNPs) (Moyerbrailean et al., 2016a), and SNPs associated with complex traits in GWAS(Pickrell, 2014). Negative controls that were not predicted to have a regulatory effect were also included in the library (Moyerbrailean et al., 2016a). It is important to note that these negative controls are only predicted not to have a regulatory effect via computation annotation (Moyerbrailean et al., 2016b), so they may not be representative of true negative controls. This is why we largely do not utilize these SNPs as negative controls within our analyses. Our predictions of regulatory activity also did not account for environmental context, thus these sequences are also not suited to annotate our cASE results.

SNP (n = 43,556): Refers to a genetic variant tested for allelic effects on gene regulation. Target (n = 43,556): 200 nucleotide-long oligonucleotide sequence that contains the test SNP in the middle of the target.

Construct (n = 87,112): Synthesized 200 nucleotide-long oligonucleotide sequence that contains only one of the two possible alleles at the test SNP. Each target corresponds to two constructs.. Direction: Constructs can integrate in either the forward or reverse direction relative to the direction of transcription in the BiT-STARR-seq plasmid. Therefore two directions are possible for each construct.

SNP/direction pair (n = 87,112): A SNP tested for allelic effects on gene regulation contrasting the expression of two constructs that are integrated in the same direction. All statistical tests are performed at this level, testing in each direction separately.”

Additionally, we now utilize the NULL sequences to annotate our ASE results, and show a depletion for these sequences. See Figure 3A. We have also prepared a plot zoomed-in to show the deviation in the 0-1 region (Author response image 2).

Figure 2 ambiguities: Thanks for pointing out this inconsistency in Supplemental figure 2. We had used the wrong caption by mistake and have now corrected it.Methods: We have added a brief description of pTWAS in the Results section to aid the reader in interpreting the results we present. Since we added additional analyses (INTACT), we have expanded our methods section to include this as well. We have also generally added more detail in the methods section.

2. Show a more convincing locus than the PIP4K2B example, which is only supported by a pTWAS result (with LD contamination issues) with no fine mapping or even nominal p-value evidence.

We agree that a stronger example could be provided. We have utilized INTACT, which identifies putative causal genes, to provide better example variants. This method combines TWAS and colocalization approaches to mitigate issues with LD contamination. We now provide the following 2 examples in the text, and include 8 total examples in supplemental table 6:

rs4938344 is an eQTL regulating the long non-coding RNA *AP000892.6*. The reference allele at this locus, G, results in decreased expression of *AP000892.6*. (as measured in GTEx and in the caffeine condition of our assay, Figure 5D and 5E respectively). INTACT associated high expression of *AP000892.6* with decreased risk of hypertension and CAD (Figure 5F). This SNP is predicted to modulate binding of GABP (a known repressor of transcription(Genuario and Perry, 1996)) and ETV1 at this site (Figure 5C). These transcription factors are upregulated in caffeine exposed endothelial cells (Findley et al., 2019) (Figure 5B). This increase in expression uncovers allelic differences in gene regulation which are not detected in the absence of caffeine, likely because of the low expression of the repressor. The allelic differences in binding of these factors should lead to allelic differences in expression of *AP000892.6*. Accordingly, the reference allele for this variant exhibited lower activity in response to caffeine in our BiT-STARR-seq experiments (Figure 5E). This effect is consistent with the GTEx artery eQTL for *AP000892.6* (Figure 5D). In summary, caffeine induces higher expression of the ETV1 and GABP transcription factors, which then bind preferentially to the reference allele at rs4938344, this results in lower expression of *AP000892.6* and increased risk for CAD and hypertension (Figure 5A). *AP000892.6* interacts with the *RB1CC1* RNA (Gong et al., 2018), which may play a role in atherosclerosis via its function in forming the autophagosome (Chen et al., 2021).

rs4527034 is an eQTL regulating the *KAT8* gene. The reference allele at this locus, A, results in decreased expression of *KAT8* (as measured in GTEx and in the control condition of our assay, Figure 6D and 6E respectively). INTACT associated high expression of *KAT8* with increased risk of hypertension (Figure 6F). This SNP is predicted to modulate binding of the TERF2IP transcription factor at this site (Figure 6C). TERF2IP is upregulated in caffeine exposed endothelial cells (Findley et al., 2019) (Figure 6B). This increase in expression may saturate all binding sites in the caffeine condition, while the transcription factor may only bind to the preferential allele in the control condition. The allelic differences in binding of these factors should lead to allelic differences in expression of *KAT8* in the control condition, which is what we observe both in our BiT-STARR-seq experiments (Figure 6E) and in GTEx artery eQTL for *KAT8* (Figure 6D). In summary, in the absence of caffeine, TERF2IP binds preferentially to the reference allele at rs4527034, this results in lower expression of *KAT8* and reduced risk for hypertension. In the presence of caffeine, TERF2IP is upregulated, resulting in increased binding and lower expression of KAT8, independently of the genotype, with an expected overall protective effect on hypertension. Confirming this potential mechanism for disease risk, TERF2IP expression levels were found to affect plaque formation in a mouse model (Kotla et al., 2019). High expression of KAT8, a histone acetyltransferase, also coincides with atherosclerotic progression, and histone acetylation increases in plaques within vascular endothelial cells (Greißel et al., 2016; Zhang et al., 2018).